# Multi-Resolution Cascades for Multiclass Object Detection

**Mohammad Saberian**
Yahoo! Labs
saberian@yahoo-inc.com

**Nuno Vasconcelos**
Statistical Visual Computing Laboratory
University of California, San Diego
nuno@ucsd.edu

## Abstract

An algorithm for learning fast multiclass object detection cascades is introduced. It produces multi-resolution (MRes) cascades, whose early stages are binary target vs. non-target detectors that eliminate false positives, late stages multiclass classifiers that finely discriminate target classes, and middle stages have intermediate numbers of classes, determined in a data-driven manner. This MRes structure is achieved with a new structurally biased boosting algorithm (SBBoost). SBBost extends previous multiclass boosting approaches, whose boosting mechanisms are shown to implement two complementary data-driven biases: 1) the standard bias towards examples difficult to classify, and 2) a bias towards difficult classes. It is shown that structural biases can be implemented by generalizing this class-based bias, so as to encourage the desired MRes structure. This is accomplished through a generalized definition of multiclass margin, which includes a set of bias parameters. SBBoost is a boosting algorithm for maximization of this margin. It can also be interpreted as standard multiclass boosting algorithm augmented with margin thresholds or a cost-sensitive boosting algorithm with costs defined by the bias parameters. A stage adaptive bias policy is then introduced to determine bias parameters in a data driven manner. This is shown to produce MRes cascades that have high detection rate and are computationally efficient. Experiments on multiclass object detection show improved performance over previous solutions.

## 1 Introduction

There are many learning problems where classifiers must make accurate decisions quickly. A prominent example is the problem of object detection in computer vision, where a sliding window is scanned throughout an image, generating hundreds of thousands of image sub-windows. A classifier must then decide if each sub-window contains certain target objects, ideally at video frame-rates, i.e. less than a micro second per window. The problem of simultaneous real-time detection of multiple class of objects subsumes various important applications in computer vision alone. These range from the literal detection of many objects (e.g. an automotive vision system that must detect cars, pedestrians, traffic signs), to the detection of objects at multiple semantic resolutions (e.g. a camera that can both detect faces and recognize certain users), to the detection of different aspects of the same object (e.g. by defining classes as different poses). A popular architecture for real-time object detection is the detector cascade of Figure 1-a [17]. This is implemented as a sequence of simple to complex classification stages, each of which can either reject the example $x$ to classify or pass it to the next stage. An example that reaches the end of the cascade is classified as a target. Since targets constitute a very small portion of the space of image sub-windows, most examples can be rejected in the early cascade stages, by classifiers of very small computation. In result, the average computation per image is small, and the cascaded detector is very fast. While the design of cascades for real-time detection of a single object class has been the subject of extensive research [18, 20, 2, 15, 1, 12, 14], the simultaneous detection of multiple objects has received much less attention.

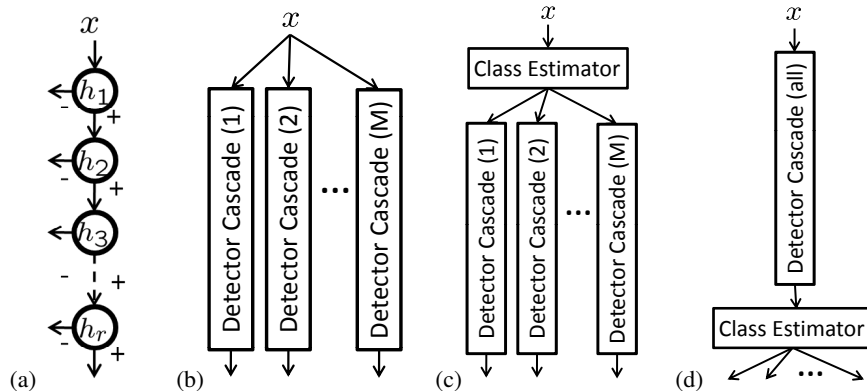

**Figure 1:** a) detector cascade [17], b) parallel cascade [19], c) parallel cascade with pre-estimator [5] and d) all-class cascade with post-estimator.

Most solutions for multiclass cascade learning simply decompose the problem into several binary (single class) detection sub-problems. They can be grouped into two main classes. Methods in the first class, here denoted *parallel cascades* [19], learn a cascaded detector per object class (e.g. view), as shown in Figure 1-b, and rely on some post-processing to combine their decisions. This has two limitations. The first is the well known sub-optimality of one-vs.-all multiclass classification, since scores of independently trained detectors are not necessarily comparable [10]. Second, because there is no sharing of features across detectors, the overall classifier performs redundant computations and tends to be very slow. This has motivated work in feature sharing. Examples include JointBoost [16], which exhaustively searches for features to be shared between classes, and [11], which implicitly partitions positive examples and performs a joint search for the best partition and features. These methods have large training complexity. The complexity of the parallel architecture can also be reduced by first making a rough guess of the target class and then running only one of the binary detectors, as in Figure 1-c. We refer to these methods as *parallel cascades with pre-estimator* [5]. While, for some applications (e.g. where classes are object poses), it is possible to obtain a reasonable pre-estimate of the target class, pre-estimation errors are difficult to undo. Hence, this classifier must be fairly accurate. Since it must also be fast, this approach boils down to real-time multiclass classification, i.e. the original problem. [4] proposed a variant of this method, where multiple detectors are run after the pre-estimate. This improves accuracy but increases complexity.

In this work, we pursue an alternative strategy, inspired by Figure 1-d. Target classes are first grouped into an abstract class of *positive patches*. A detector cascade is then trained to distinguish these patches from everything else. A patch identified as positive is finally fed to a multiclass classifier, for assignment to one of the target classes. In comparison to parallel cascades, this has the advantage of sharing features across all classes, eliminating redundant computation. When compared to the parallel cascade with pre-estimator, it has the advantage that the complexity of its class estimator has little weight in the overall computation, since it only processes a small percentage of the examples. This allows the use of very accurate/complex estimators. The main limitation is that the design of a cascade to detect all positive patches can be quite difficult, due to the large intra-class variability. This is, however, due to the abrupt transition between the all-class and multiclass regimes. While it is difficult to build an all-class detector with high detection and low false-positive rate, we show that this is really not needed. Rather than the abrupt transition of Figure 1-d, we propose to learn a multiclass cascade that gradually progresses from all-class to multiclass. Early stages are binary all-class detectors, aimed at eliminating sub-windows in background image regions. Intermediate stages are classifiers with intermediate numbers of classes, determined by the structure of the data itself. Late stages are multiclass classifiers of high accuracy/complexity. Since these cascades represent the set of classes at different resolutions, they are denoted multi-resolution (MRes) cascades.

To learn MRes cascades, we consider a $M$-class classification problem and define a negative class $M + 1$, which contains all non-target examples. We then analyze a recent multiclass boosting algorithm, MCBoost [13], showing that its weighting mechanism has two components. The first is the standard weighting of examples by how well they are classified at each iteration. The second, and more relevant to this work, is a similar weighting of the classes according to their difficulty. MC-

Boost is shown to select the weak learner of largest margin on the reweighted training sample, under a *biased* definition of margin that reflects the class weights. This is a data-driven bias, based purely on classification performance, which does not take computational efficiency into account. To induce the MRes behavior, it must be complemented by a *structural bias* that modifies the class weighting to encourage the desired multi resolution structure. We show that this can be implemented by augmenting MCBoost with structural bias parameters that lead to a new *structurally biased boosting algorithm* (SBBoost). This can also be seen as a variant of boosting with tunable margin thresholds or as boosting under a cost-sensitive risk. By establishing a connection between the bias parameters and the computational complexity of cascade stages, we then derive a *stage adaptive bias policy* that guarantees computationally efficient MRes cascades of high detection rate. Experiments in multi-view car detection and simultaneous detection of multiple traffic signs show that the resulting classifiers are faster and more accurate than those previously available.

## 2   Boosting with structural biases

Consider the design of a $M$ class cascade. The $M$ target classes are augmented with a class $M + 1$, the *negative class*, containing non-target examples. The goal is to learn a multiclass cascade detector $\mathcal{H}[h_1(x), \ldots, h_r(x)]$ with $r$ stages. This has the structure of Figure 1-a but, instead of a binary detector, each stage is a multiclass classifier $h_k(x) : \mathcal{X} \rightarrow \{1, \ldots, M + 1\}$. Mathematically,

$$\mathcal{H}[h_1(x), \ldots, h_r(x)] = \begin{cases} h_r(x) & \text{if } h_k(x) \neq M + 1 \; \forall k, \\ M + 1 & \text{if } \exists k | \; h_k(x) = M + 1. \end{cases} \tag{1}$$

We propose to learn the cascade stages with an extension of the MCBoost framework for multiclass boosting of [13]. The class labels $\{1, \ldots, M + 1\}$ are first translated into a set of codewords $\{y_1, \ldots, y_{M+1}\} \in \mathbb{R}^M$ that form a simplex where $\sum_{i=1}^{M+1} y_i = 0$. MCBoost uses the codewords to learn a $M$-dimensional predictor $F^*(x) = [f_1(x), \ldots, f_M(x)] \in \mathbb{R}^M$ so that

$$\begin{cases} F^*(x) = \arg\min_{F(x)} & \overline{\mathcal{R}}[F] = \frac{1}{n} \sum_{i=1}^{n} \sum_{\substack{j=1}}^{M+1} e^{-\frac{1}{2}[\langle y_{z_i}, F(x_i) \rangle - \langle y_j, F(x_i) \rangle]} \\ s.t & F(x) \in span(\mathcal{G}), \end{cases} \tag{2}$$

where $\mathcal{G} = \{g_i\}$ is a set of weak learners. This is done by iterative descent [3, 9]. At each iteration, the best update for $F(x)$ is identified as

$$g_k^* = \arg\max_{g \in \mathcal{G}} -\delta\overline{\mathcal{R}}[F; g], \tag{3}$$

with

$$-\delta\overline{\mathcal{R}}[F; g] = - \left. \frac{\partial\overline{\mathcal{R}}[f^t + \epsilon g]}{\partial\epsilon} \right|_{\epsilon=0} = \frac{1}{2} \sum_{i=1}^{n} \sum_{k=1}^{M+1} \langle g(x_i), y_{z_i} - y_k \rangle e^{-\frac{1}{2}\langle F(x_i), y_{z_i} - y_k \rangle}. \tag{4}$$

The optimal step size along this weak learner direction is

$$\alpha^* = \arg\min_{\alpha \in \mathbb{R}} R[F(x) + \alpha g^*(x)], \tag{5}$$

and the predictor is updated according to $F(x) = F(x) + \alpha^* g^*(x)$. The final decision rule is

$$h(x) = \arg\max_{k=1 \ldots M+1} \langle y_k, F^*(x_i) \rangle. \tag{6}$$

We next provide an analysis of the updates of (3) which inspires the design of MRes cascades.

**Weak learner selection:** the multiclass margin of predictor $F(x)$ for an example $x$ from class $z$ is

$$\mathcal{M}(z, F(x)) = \langle F(x), y_z \rangle - \max_{j \neq z}\langle F(x), y_j \rangle = \min_{j \neq z}\langle F(x), y_z - y_j \rangle, \tag{7}$$

where $\langle F(x), y_z - y_j \rangle$ is the *margin component* of $F(x)$ with respect to class $j$. Rewriting (3) as

$$-\delta R[F; g] = \frac{1}{2} \sum_{i=1}^{n} \sum_{k=1|k \neq z_i}^{M+1} \langle g(x_i), y_{z_i} - y_k \rangle e^{-\frac{1}{2}\langle F(x_i), y_{z_i} - y_k \rangle} \tag{8}$$

$$= \frac{1}{2} \sum_{i=1}^{n} w(x_i) \langle g(x_i), \sum_{k=1|k \neq z_i}^{M+1} \rho_k(x_i)(y_{z_i} - y_k) \rangle, \tag{9}$$

where

$$w(x_i) = \sum_{k=1|k \neq z_i}^{M} e^{-\frac{1}{2}\langle F(x_i), y_{z_i} - y_k \rangle}, \quad \rho_k(x_i) = \frac{e^{-\frac{1}{2}\langle F(x_i), y_{z_i} - y_k \rangle}}{\sum_{k=1|k \neq z_i}^{M} e^{-\frac{1}{2}\langle F(x_i), y_{z_i} - y_k \rangle}}. \tag{10}$$

enables the interpretation of MCBoost as a generalization of AdaBoost. From (10), an example $x_i$ has large weight $w(x_i)$ if $F(x_i)$ has at least one large negative margin component, namely

$$\langle F(x_i), y_z - \overline{y} \rangle < 0 \qquad \text{for} \qquad \overline{y} = \arg\min_{y_j \neq y_z} \langle F(x_i), y_z - y_j \rangle. \tag{11}$$

In this case, it follows from (6) that $x_i$ is incorrectly classified into the class of codeword $\overline{y}$. In summary, as in AdaBoost, the weighting mechanism of (9) emphasizes examples incorrectly classified by the current predictor $F(x)$. However, in the multiclass setting, this is only part of the weighting mechanism, since the terms $\rho_k(x_i)$ of (9)-(10) are coefficients of a soft-min operator over margin components $\langle F(x_i), y_{z_i} - y_k \rangle$. Assuming the soft-min closely approximates the min, (9) becomes

$$-\delta R[F; g] \approx \sum_{i=1}^{n} w(x_i) \mathcal{M}_F(y_{z_i}, g(x_i)), \tag{12}$$

where

$$\mathcal{M}_F(z, g(x)) = \langle g(x), y_z - \overline{y} \rangle. \tag{13}$$

and $\overline{y}$ is the codeword of (11). This is the multiclass margin of weak learner $g(x)$ under an alternative margin definition $\mathcal{M}_F(z, g(x))$. Comparing to the original definition of (7), which can be written as

$$\mathcal{M}(z, g(x)) = \frac{1}{2}\langle g(x), y_z - \overline{\overline{y}} \rangle \qquad \text{where} \qquad \overline{\overline{y}} = \arg\min_{y_j \neq y_z} \langle g(x), y_z - y_j \rangle, \tag{14}$$

$\mathcal{M}_F(y_z, g(x))$ restricts the margin of $g(x)$ to the worst case codeword $\overline{y}$ for the current predictor $F(x)$. The strength of this restriction is determined by the soft-min operator. If $< F(x), y_z - \overline{y} >$ is much smaller than $< F(x), y_z - y_j > \; \forall y^j \neq \overline{y}$, $\rho_k(x)$ closely approximates the minimum operator and (12) is identical to (9). Otherwise, the remaining codewords also contribute to (9). In summary, $\rho_k(x_i)$ is a set of class weights that emphasizes classes of small margin for $F(x)$. The inner product of (9) is *the margin of $g(x)$ after this class reweighting*. Overall, MCBoost weights introduce a bias towards difficult examples (weights $w$) *and* difficult classes (margin $\mathcal{M}_F$).

**Structural biases:** The core idea of cascade design is to bias the learning algorithm towards computationally efficient classifier architectures. This is not a data driven bias, as in the previous section, but a *structural bias,* akin to the use of a prior (in Bayesian learning) to guarantee that a graphical model has a certain structure. For example, because classifier speed depends critically on the ability to quickly eliminate negative examples, the initial cascade stages should effectively behave as a binary classifier (all classes vs. negative). This implies that the learning algorithm should be biased towards predictors of large margin component $\langle F(x), y_z - y_{M+1} \rangle$ with respect to the negative class $j = M + 1$. We propose to implement this structural bias by forcing $y_{M+1}$ to be the dominant codeword in the soft-min weighting of (10). This is achieved by rescaling the soft-min coefficients, i.e. by using an alternative soft-min operator $\rho_k^\alpha(x_i) \propto \alpha_k e^{-\frac{1}{2}\langle F(x_i), y_{z_i} - y_k \rangle}$, where $\alpha_k = \tau \in [0, 1]$ for $k \neq M + 1$ and $\alpha_{M+1} = 1$. The parameter $\tau$ controls the strength of the structural bias. When $\tau = 0$, $\rho_k^\alpha(x_i)$ assigns all weight to codeword $y_{M+1}$ and the structural bias dominates. For $0 < \tau < 1$ the bias of $\rho_k^\alpha(x_i)$ varies between the data driven bias of $\rho_k(x_i)$ and the structural bias towards $y_{M+1}$. When $\tau = 1$, $\rho_k^\alpha(x_i) = \rho_k(x_i)$, the bias is purely data driven, as in MCBoost. More generally, we can define biases towards any classes (beyond $j = M + 1$) by allowing different $\alpha_k \in [0, 1]$ for different $k \neq M + 1$. From (10), this is equivalent to redefining the margin components as $\langle F(x_i), y_{z_i} - y_k \rangle - 2\log\alpha_k$. Finally the biases can be adaptive with respect to the class of $x_i$, by redefining the margin components as $\langle F(x_i), y_{z_i} - y_k \rangle - \delta_{z_i,k}$. Under this *structurally biased* margin, the approximate boosting updates of (12) become

$$-\delta R[F; g] \approx \sum_{i=1}^{n} w(x_i) \mathcal{M}_F^c(y_{z_i}, g(x_i)), \tag{15}$$

where

$$\mathcal{M}_F^c(z, g(x)) = \langle g(x), y_z - \hat{y} \rangle - \delta_{z_i,k} \qquad \hat{y} = \arg\min_{y_j \neq y_z} \langle F(x), y_z - y_j \rangle - \delta_{z_i,k}. \tag{16}$$

This is, in turn, equivalent to the approximation of (9) by (12) under the definition of margin as

$$\mathcal{M}^c(z, F(x)) = \min_{j \neq z} \langle F(x), y_z - y_j \rangle - \delta_{z,j}, \tag{17}$$

and boosting weights

$$w^c(x_i) = \sum_{k=1|k \neq z_i}^{M} e^{-\frac{1}{2}[\langle F(x_i), y_{z_i} - y_k \rangle - \delta_{z_i,k}]}, \quad \rho_k^c(x_i) = \frac{e^{-\frac{1}{2}[\langle F(x_i), y_{z_i} - y_k \rangle - \delta_{z_i,k}]}}{\sum_{l=1|l \neq z_i}^{M} e^{-\frac{1}{2}[\langle F(x_i), y_{z_i} - y_l \rangle - \delta_{z_i,l}]}}. \tag{18}$$

We denote the boosting algorithm with these weights as *structurally biased* boosting (SBBoost).

**Alternative interpretations:** the parameters $\delta_{z_i,k}$, which control the amount of structural bias, can be seen as thresholds on the margin components. For binary classification, where $M = 1, y_1 = 1, y_2 = -1$ and $F(x)$ is scalar, (7) reduces to the standard margin $\mathcal{M}(z, F(x)) = y_z F(x)$, (10) to the standard boosting weights $w(x_i) = e^{-y_{z_i} F(x_i)}$ and $\rho_k(x_i) = 1, k \in \{1, 2\}$. In this case, MC-Boost is identical to AdaBoost. SBBoost can thus been seen as an extension of AdaBoost, where the margin is redefined to include thresholds $\delta_{z_i}$ according to $\mathcal{M}^c(z, F(x)) = y_z F(x) - \delta_z$. By controlling the thresholds it is possible to bias the learned classifier towards accepting or rejecting more examples. For multiclass classification, a larger $\delta_{z,j}$ encodes a larger bias against assigning examples from class $z$ to class $j$. This behavior is frequently denoted as *cost-sensitive* classification. While it can be achieved by training a classifier with AdaBoost (or MCBoost) and adding thresholds to the final decision rule, this is suboptimal since it corresponds to using a classification boundary on which the predictor $F(x)$ was not trained [8]. Due to Boosting's weighting mechanism (which emphasizes a small neighborhood of the classification boundary), classification accuracy can be quite poor when the thresholds are introduced a-posteriori. Significantly superior performance is achieved when the thresholds are accounted for by the learning algorithm, as is the case for SBBoost. Boosting algorithms with this property are usually denoted as cost-sensitive and derived by introducing a set of classification costs in the risk of (2). It can be shown, through a derivation identical to that of Section 2, that SBBoost is a cost-sensitive boosting algorithm with respect to the risk

$$\overline{\mathcal{R}}^c[F] = \frac{1}{n} \sum_{i=1}^{n} \sum_{j=1}^{M+1} C_{z,j} e^{-\frac{1}{2}\langle y_{z_i}, F(x_i) \rangle - \langle y_j, F(x_i) \rangle} \tag{19}$$

with $\delta_{z,j} = \frac{1}{2} \log C_{z,j}$. Under this interpretation, the bias parameters $\delta_{z,j}$ are the log-costs of assigning examples of class $z$ to class $j$. For binary classification, SBBoost reduces to the cost-sensitive boosting algorithm of [18].

## 3 Boosting MRes cascades

In this section we discuss a strategy for the selection of bias parameters $\delta_{i,j}$ that encourage multi-resolution behavior. We start by noting that some biases must be shared by all stages. For example, while a cascade cannot recover a rejected target, the false-positives of a stage can be rejected by its successors. Hence, the learning of each stage must enforce a bias against target rejections, at the cost of increased false-positive rates. This *high detection rate problem* has been the subject of extensive research in binary cascade learning, where a bias against assigning examples to the negative class is commonly used [18, 8]. The natural multiclass extension is to use much larger thresholds for the margin components with respect to the negative class than the others, i.e.

$$\delta_{k,M+1} \gg \delta_{M+1,k} \,\forall k = 1, \ldots, M. \tag{20}$$

We implement this bias with the thresholds

$$\delta_{k,M+1} = \log \beta \qquad \delta_{M+1,k} = \log(1 - \beta) \qquad \beta \in [0.5, 1]. \tag{21}$$

The value of $\beta$ is determined by the target detection rate of the cascade. For each boosting iteration, we set $\beta = 0.5$ and measure the detection rate of the cascade. If this falls below the target rate, $\beta$ is increased to $(\beta + 1)/2$. The process is repeated until the desired rate is achieved.

There is also a need for structural biases that vary with the cascade stage. For example, the computational complexity $c^{t+1}$ of stage $t + 1$ is proportional to the product of the per-example complexity

$\epsilon^{t+1}$ of the classifier (e.g. number of weak learners) and the number of image sub-windows that it evaluates. Since the latter is dominated by the false positives rate of the previous cascade stages, $fp^t$, it follows that $c^{t+1} \propto fp^t \epsilon^{t+1}$. Since $fp^t$ decreases with $t$, an efficient cascade must have early stages of low complexity and more complicated detectors in later stages. This suggests the use of stages that gradually progress from binary to multiclass. Early stages eliminate false-positives, late stages are accurate multiclass classifiers. In between, the cascade stages should detect intermediate numbers of classes, according to the structure of the data. Cascades with this structure represent the set of classes at different resolutions and are denoted Multi-Resolution (MRes) cascades.

To encourage the MRes structure, we propose the following *stage adaptive bias policy*

$$\delta_{k,l}^t = \begin{cases} \gamma^t = \log \frac{FP}{fp^t} & \forall k,l \in \{1,\dots,M\} \\ \log \beta & \text{for } k \in \{1,\dots,M\} \text{ and } l = M+1 \\ \log(1-\beta) & \text{for } k = M+1 \text{ and } l \in \{1,\dots,M\}, \end{cases} \tag{22}$$

where $FP$ is the target false-positive rate for the whole cascade. This policy complements the stage-independent bias towards high detection rate (due to $\beta$) with a stage dependent bias $\delta_{k,l}^t = \gamma^t, \forall k,l \in \{1,\dots,M\}$. This has the following consequences. First, since $\beta \geq 0.5$ and $fp^t \gg 2FP$ when $t$ is small, it follows that $\gamma^t \ll \delta_{k,M+1}$ in the early stages. Hence, for these stages, there is a much larger bias against rejection of examples from the target classes $\{1,\dots,M\}$, than for the differentiation of these classes. In result, the classifier $h_t(x)$ is an all-class detector, as in Figure 1-d. Second, for large $t$, where $fp^t$ approaches FP, $\gamma^t$ decreases to zero. In this case, there is no bias against class differentiation and the learning algorithm places less emphasis on improvements of false-positive rate ($\delta_{k,M+1} \approx \gamma^t$) and more emphasis on target differentiation. Like MCBoost (which has no biases), it will focus in the precise assignment of targets to their individual classes. In result, for late cascade stages, $h_t(x)$ is a multiclass classifier, similar to the class post-estimator of Figure 1-d. Third, for intermediate $t$, it follows from (19) and $e^{\gamma^t} \propto \epsilon^{t+1}/c^{t+1}$ that the learned cascade stages are optimal under a risk with costs $C_{z,j}^t \propto 1/\nu^{t+1}$, for $z,j \in \{1,\dots,M\}$ where $\nu^t = c^t/\epsilon^t$. Note that $\nu^t$ is a measure of how much the computational cost per example is magnified by stage $t$, therefore this risk favors cascades with stages of *low complexity magnification*. In result, weak learners are preferentially added to the stages where their addition produces the smallest overall computational increase. This makes the resulting cascades computationally efficient, since 1) stages of high complexity magnification have small per example complexity $\epsilon^t$ and 2) classifiers of large per example complexity are pushed to the stages of low complexity magnification. Since complexity magnification is proportional to false-positive rate ($c^t/\epsilon^t \propto fp^{t-1}$), multiclass decisions (higher $\epsilon^t$) are pushed to the latter cascade stages. This push is data driven and gradual and thus the cascade gradually transitions from binary to multiclass, becoming a soft version of the detector of Figure 1-d.

## 4 Experiments

SBBoost was evaluated on the tasks of multi-view car detection, and multiple traffic sign detection. The resulting MRes cascades were compared to the detectors of Figure 1. Since it has been established in the literature that the all-class detector with post-estimation has poor performance [5], the comparison was limited to parallel cascades [19] and parallel cascades with pre-estimation [5]. All binary cascade detectors were learned with a combination of the ECBoost algorithm of [14] and the cost-sensitive Boosting method of [18]. Following [2], all cascaded detectors used integral channel features and trees of depth two as weak learners. The training parameters were set to $\eta = 0.02$, $D = 0.95$, $FP = 10^{-6}$ and the training set was bootstrapped whenever the false positive rate dropped below 90%. Bootstrapping also produced an estimate of the real false positive rate $fp^t$, used to define the biases $\delta_{k,l}^t$. As in [5], the detector cascade with pre-class estimation used tree classifiers for pre-estimation. In the remainder of this section, detection rate is defined as the percentage of target examples, from all views or target classes, that were detected. Detector accuracy is the percentage of the target examples that were detected and assigned to the correct class. Finally, detector complexity is the average number of tree node classifiers evaluated per example.

**Multi-view Car Detection:** To train a multi-view car detector, we collected images of 128 Frontal, 100 Rear, 103 Left, and 103 Right car views. These were resized to $41 \times 70$ pixels. The multi-view car detector was evaluated on the USC car dataset [6], which consists of 197 color images of size $480 \times 640$, containing 410 instances of cars in different views.

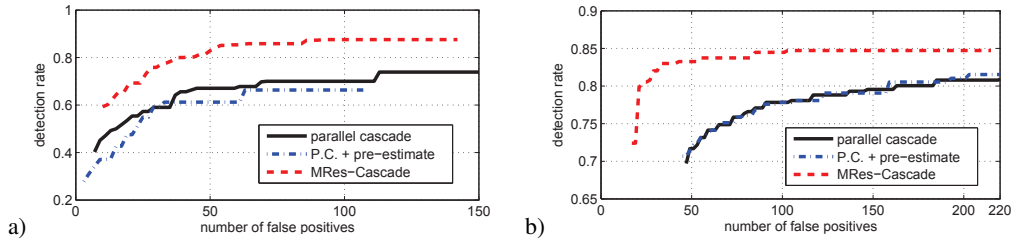

**Figure 2:** ROCs for a) multi-view car detection and b) traffic sign detection.

**Table 1:** Multi-view car detection performance at 100 false positives.

| Method | car detection | | | traffic sign detection | | |
|---|---|---|---|---|---|---|
| | *complexity* | *accuracy* | det. rate | *complexity* | *accuracy* | det. rate |
| *Parallel Cascades [19]* | 59.94 | 0.35 | 0.72 | 10.08 | 0.78 | 0.78 |
| *P.C. + Pre-estimation [5]* | $15.15 + 6$ | 0.35 | 0.70 | $2.32 + 4$ | 0.78 | 0.78 |
| *MRes cascade* | **16.40** | **0.58** | **0.88** | **5.56** | **0.84** | **0.84** |

The ROCs of the various cascades are shown in Figure 2-a. Their detection rate, accuracy and complexity are reported in Table 1. The complexity of parallel cascades with pre-processing is broken up into the complexity of the cascade plus the complexity of the pre-estimator. Figure 2-a, shows that the MRes cascade has significantly better ROC performance than any of the other detectors. This is partially due to the fact that the detector is learned jointly across classes and thus has access to more training examples. In result, there is less over-fitting and better generalization. Furthermore, as shown in Table 1, the MRes cascade is much faster. The 3.5-fold reduction of complexity over the parallel cascade suggests that MRes cascades share features very efficiently across classes. The MRes cascade also detects $16\%$ more cars and assigns $23\%$ more cars to the true class. The parallel cascade with pre-processing was slightly less accurate than the parallel cascade but three times as fast. Its accuracy is still $23\%$ lower than that of the MRes cascade and the complexity of the pre-estimator makes it $20\%$ slower.

Figure 3 shows the evolution of the detection rate, false positive rate, and accuracy of the MRes cascade with learning iterations. Note that the detection rate is above the specified $D = 95\%$ throughout the learning process. This is due to the updating of the $\beta$ parameter of (22). It can also be seen that, while the false positive rate decreases gradually, accuracy remains low for many iterations. This shows that the early stages of the MRes cascade place more emphasis on rejecting negative examples (lowering the false positive rate) than making precise view assignments for the car examples. This reflects the structural biases imposed by the policy of (22). Early on, confusion between classes has little cost. However, as the cascade grows and its false positive rate $fp^t$ decreases, the detector starts to distinguish different car views. This happens soon after iteration 100, where there is a significant jump in accuracy. Note, however, that the false-positive rate is still $10^{-4}$ at this point. In the remaining iterations, the learning algorithm continues to improve this rate, but also "goes to work" on increasing accuracy. Eventually, the false-positive rate flattens and the SBBoost behaves as a multiclass boosting algorithm. Overall, the MRes cascade behaves as a soft version of the all-class detector cascade with post-estimation, shown in Figure 1-d.

**Traffic Sign Detection:** For the detection of traffic signs, we extracted $1,159$ training examples from the first set of the Summer traffic sign dataset [7]. This produced 660 examples of "priority road", 145 of "pedestrian crossing", 232 of "give way" and 122 of "no stopping no standing" signs. For training, these images were resized to $40 \times 40$. For testing, we used 357 images from the second set of the Summer dataset which contained at least one visible instance of the traffic signs, with more than 35 pixels of height. The performance of different traffic sign detectors is reported in Figure 2-b) and Table 1. Again, the MRes cascade was faster and more accurate than the others. In particular, it was faster than other methods, while detecting/recognizing $6\%$ more traffic signs.

We next trained a MRes cascade for detection of the 17 traffic signs shown in the left end of Figure 4. The figure also shows the evolution of MRes cascade decisions for 20 examples from each of the different classes. Each row of color pixels illustrates the evolution of one example. The color of the $k^{th}$ pixel in a row indicates the decision made by the cascade after $k$ weak learners. The traffic signs and corresponding colors are shown in the left of the figure. Note that the early cascade stages only reject a few examples, assigning most of the remaining to the first class. This assures

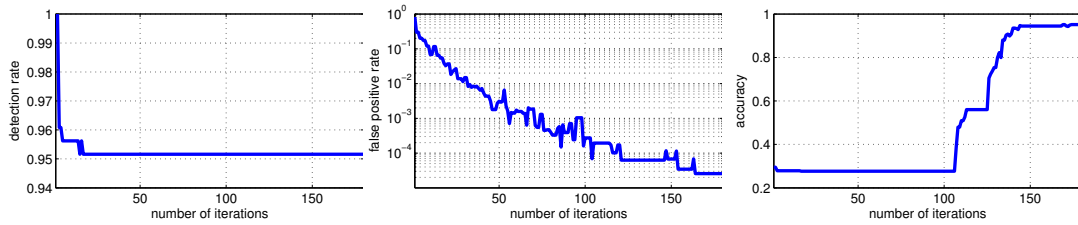

**Figure 3:** MRes cascade detection rate (left), false positive rate (center), and accuracy (right) during learning.

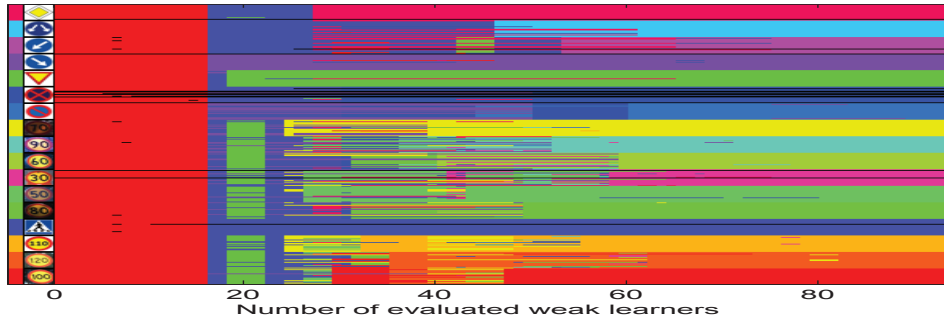

**Figure 4:** Evolution of MRes cascade decisions for 20 randomly selected examples of 17 traffic sign classes. Each row illustrates the evolution of the label assigned to one example. The ground-truth traffic sign classes and corresponding label colors are shown on the left.

a high detection rate but very low accuracy. However, as more weak learners are evaluated, the detector starts to create some intermediate categories. For example, after 20 weak learners, all traffic signs containing red and yellow colors are assigned to the "give way" class. Evaluating more weak learners further separates these classes. Eventually, almost all examples are assigned to the correct class (right side of the picture). This shows that besides being a soft version of the all-class detector cascade, the MRes cascade automatically creates an internal class taxonomy.

Finally, although we have not produced detection ground truth for this experiment, we have empirically observed that the final 17-traffic sign MRes cascade is accurate and has low complexity (5.15). This make it possible to use the detector in real-time on low complexity devices, such as smart-phones. A video illustrating the detection results is available in the supplementary material.

## 5 Conclusion

In this work, we have made various contributions to multiclass boosting with structural constraints and cascaded detector design. First, we proposed that a multiclass detector cascade should have MRes structure, where early stages are binary target vs. non-target detectors and late stages perform fine target discrimination. Learning such cascades requires the addition of a structural bias to the learning algorithm. Second, to incorporate such biases in boosting, we analyzed the recent MC-Boost algorithm, showing that it implements two complementary weighting mechanisms. The first is the standard weighting of examples by difficulty of classification. The second is a redefinition of the margin so as to weight more heavily the most difficult classes. This class reweighting was interpreted as a data driven class bias, aimed at optimizing classification performance. This suggested a natural way to add structural biases, by modifying class weights so as to favor the desired MRes structure. Third, we showed that such biases can be implemented through the addition of a set of thresholds, the bias parameters, to the definition of multiclass margin. This was, in turn, shown identical to a cost-sensitive multiclass boosting algorithm, using bias parameters as log-costs of mis-classifying examples between pairs of classes. Fourth, we introduced a stage adaptive policy for the determination of bias parameters, which was shown to enforce a bias towards cascade stages of 1) high detection rate, and 2) MRes structure. Cascades designed under this policy were shown to have stages that progress from binary to multiclass in a gradual manner that is data-driven and computationally efficient. Finally, these properties were illustrated in fast multiclass object detection experiments involving multi-view car detection and detection of multiple traffic signs. These experiments showed that MRes cascades are faster and more accurate than previous solutions.

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
