[Reviews · NeurIPS 2014]

Submitted by Assigned_Reviewer_17

This paper proposed a multi-resolution (MRes) cascades method for multiclass object detection. By redefining multiclass margin and implementing the structural biases, this method can force the behavior of the detector at different stage. At early stages, the multiclass classifiers are binary detectors aiming at deleting false positives quickly. In middle stages, the classifiers progress gradually from binary to multiclass by modifying class weights. In final stages, the multiclass classifiers perform fine target discrimination to improve the detection accuracy.

This structure is very intuitive and computationally efficient. It performs better than both parallel cascade and parallel cascades with pre-estimator. However, there are several drawbacks of this paper.

(1) The two datasets used in this paper are generated with 5 years. But the baseline methods compared with are more than 10 years ago. It would be more convincing to compare with more recent methods on multiclass object detection.

(2) The detection rate of USC car dataset in the original paper [4] is 0.985, which is much better than what is proposed in this paper 0.88. So has this proposed method improved the state of the art?

(3) For Fig. 3, it seems that there are two jumps in accuracy during training process. Is there any explanation for this phenomenon? Or is it reasonable to assume that the stages are not gradually changed from the binary to multiclass classifier?

(4) The experiment for 17 traffic signs is very interesting but lack more detailed results and comparison.
Summary: A paper with interesting idea, but the experiments are less than convincing.

Submitted by Assigned_Reviewer_24

This paper proposes a new cascading system for multiclass object detection. The algorithm exploits a new multiclass structurally biased boosting algorithm (SBBoost) that allows to introduce a bias towards some specific classes. SBBoost is used to implement a cascade where first models are biased towards the elimination of false positives (ie., the good detection of the negative class), late models are biased to finely discriminate the target classes, and middle models are biased in-between. The proposed MRes cascade is shown to outperform other cascades models on two object detection problems.

The problem of multiclass object detection is important and has many applications. Building cascades for this setting is also very interesting to minimize detection times. Although several solutions already exist for this problem, the proposed method is original in that it solves the problem in a more integrated way than existing mutliclass cascading systems. A second contribution of the paper is the introduction of a new cost-sensitive multiclass boosting algorithm, called SBBoost. This algorithm seems also original but a comparison with related works on cost-sensitive multiclass boosting is missing in the paper (which focuses on the cascade aspect).

The derivation of the SBBoosting method from the MCBoost method of Saberian and Vasconcelos is sound. It is first shown that MCBoost naturally introduces a bias towards difficult classes and this feature is exploited to introduce artificial class biases with the introduction of new bias parameters into the margin definition. Interestingly, these parameters can be interpreted as the logs of the cost of assigning examples from one class to another.

The derivation of the MRes cascade model makes also much sense. I like the idea of going progressively from target detection to the discrimination of the classes. The proposed stage adaptive bias policy should clearly achieve this goal but it appears however to be not very well motivated from a theoretical point view. It is not clear which global criterion it optimizes and whether other policies could not achieve better performance.

Experiments clearly show that the MRes cascades outperform other cascades on two different object detection problems. The compared methods are however rather complex and depends on several parameters that are not given in the paper. For this reason, while results of MRes are very good, we can only trust the authors that the other methods have been tuned appropriately for the comparison. With the information provided in the paper, it is clearly impossible to reproduce the experiments.

I’m also disappointed that the proposed method is only evaluated as a whole. There is no analysis of the effect of its parameters or evaluation of the different design choices. For example, it would be interesting to turn off the stage adaptive bias policy to see how important it is to achieve good performance. What is the performance achieved by the MCE method without adaption? The newly proposed SBBoost method is also not validated per se. It would be interesting to compare it with other cost-sensitive multiclass boosting methods. Is this method crucial in the context of the cascade or could it be replaced by another cost-sensitive approach with similar performance?

The paper is well written. It is mostly clear and well organized. While all individual steps of the algorithm are clear, it is not so clear however how all pieces are put together. It should help to have a pseudo-code description of the method. For example, while I think I understand how the cascade is trained, I’m not sure how exactly it is used to compute predictions. In Equation (1), what are the h_i(x) models? Is h_i(x) the ith weak learner or does it correspond to the sum of the predictions from all weak learners until the ith one? I’m not sure also to understand how the update of the beta is interleaved with the adaptive policy of (22).

Minor comments:
- The conclusion is not very useful as it is a simple summary of the paper without any discussion of the method or of future works.
- Page 5,  « \beta is increased to (\beta+1)/2 » -> decreased
Summary: The paper proposes a very interesting and well thought new cascading system for multiclass object detection, as well as a new cost-sensitive multiclass boosting method. The cascading system remains however rather heuristic and its empirical validation is not totally convincing.

Submitted by Assigned_Reviewer_42

The paper introduces a multi-class cascading classifier technique incorporating the concept of structural bias. The system is trained such that patches of interest are first picked out from the rest of the data, then more specialized classifiers are applied to identify categories.

Strong points:
- There is novelty in how the margin changes as the classification task needs to be more stringent (i.e. at first, errors are tolerable, however, in later stages, the learning procedure ensures they are eliminated).
- Considerable improvement in accuracy and reduction of complexity compared to the state-of-the-art in terms of cascade detectors using boosting.
- The experiment shown in Figure 4 shows how classification occurs gradually within the system, indicating that the classifier builds an intrinsic hierarchy (a behaviour sometimes observed in neural networks),

Some drawbacks:
- Techniques such as deep belief networks not considered for this task, though in practice they are often employed.
- The formatting of the paper is odd and makes it difficult to read; if accepted, please use a different font for the camera ready.

Following the author feedback, I maintain my opinion that this paper should be accepted.
Summary: The paper presents a well-performing solution for an extensively studied problem. The solution seems technically sound, with sufficiently convincing experimental results to support its usefulness.
Author Feedback
Author rebuttal: R1:
- Competing methods are old
o There has been significant work in object detection in last decade, e.g. PASCAL VOC, but little has addressed real-time simultaneous detection of multiple object classes. Most works are 1) detectors of a single object class, e.g, 1-vs-all dominates in VOC, and 2) too complex for real-time implementation on low complexity devices, e.g. embedded processors used in smart vehicles. Note that goal is to detect objects of unknown scale in a large image (the “detection task” on VOC). Most methods outside the cascade literature are not even close to real-time on this task (see reply to R3). Hence, they are not feasible solutions for the applications we consider. If the complexity constraint were released, even we could possibly do better, e.g., by using more complex weak learners. But this would prevent real-time implementation and is beyond the scope of the work.

- Detection rate of USC car dataset in [4] is 0.985, better than 0.88
o This is not true. The 0.985 rate of [4] is for single scale side-view car detection on UIUC dataset (Table 1 of [4]). For this task - only one target class (side-view only) - our method is equivalent to [13]. As shown in [13], detection rate on UIUC is 0.99. Finally note that our proposed method is not directly comparable with that of [4] since the proposed. The 0.88 rate that we reported is for the multi view (multiple target class) dataset introduced by [4]. Finally note that the proposed method of [4] is not directly comparable with our method since it only detects but our method detects and recognizes the targets.

- For Fig. 3, it seems 2 jumps in accuracy during training. Any explanation? Or can we assume that stages don’t gradually change from binary to multiclass?
o Good question. Some of this is discussed in L352-364 but we expand the answer here and will update the paper accordingly. First, for better analysis, we generated the prediction evolution plot, (similar to Fig 4) for this car experiment. We observed that the algorithm starts by grouping all targets in the “car” class and learning a 2 class detector (cars vs. non-cars). Around iteration 100, the detector groups cars into two classes (side views vs. frontal/rear view) which increases the accuracy to around 50%. Then, around iteration 150 algorithm starts to distinguish between frontal and rear view which increases the accuracy to around 75%. Finally the algorithm learns to distinguish between right-sided and left-sided cars and the accuracy gets close to 100%. So, we would say that the stages gradually change from binary to multiclass, but there is also “phase transitions” behavior which we believe is responsible for the rapid accuracy improvement in each phase. This behavior is similar to the phase transition phenomena in physics where once a matter reaches to a critical condition, e.g. 100 degree, it goes from one state to another. We believe that there is a similar mechanism in our method, i.e. once the policy reaches some critical values the detector rapidly starts learning more number of classes.

- More details on 17 traffic sign detection
o Good suggestion. We had limited space in submission, but will try to include in the paper supplement.
R2:
- What is the global criterion that adaptive bias policy optimizes?
o Excellent question. The optimal bias policy is the one that results in fastest cascade detectors. In this paper we relate this global criterion to the FP rate of the cascade after each stage (line 273) and adapt the bias accordingly. This is discussed in L269-273 and L294-303.

- Experiments not reproducible
o The code will be made available online.

- MRes evaluated as whole, individual parts not analyzed, e.g. without adaptation? Or compare to multiclass cost-sensitive Boosting algorithms
o MCBoost, the cost-insensitive boosting algorithm that inspires SBBoost, is state-of-the-art for cost-insensitive multiclass boosting [12]. Without adaptation of (22) cascade will not evolve from binary to multiclass, and real-time performance is impossible. R2 correct that (19) could be optimized by other cost-sensitive methods. But without derivation of section 2 this would be weakly motivated, so we did not think of it. We will report in paper supplement.

- pseudo-code?
o Will be added to paper supplement.

- What are h_i(x)?
o Each cascade stage has a multiclass predictor, f^r(x) = (f^r_1(x), … f^r_M(x)). These are learned with the cost-sensitive extension of CD-MCBoost [12], i.e. using weights of (18). Classifier of stage r, h_r(x), is then given by (6).

- How is update of beta interleaved with adaptive policy of (22)?
o beta and (22) are updated independently, based on detection and FP rate respectively. Beta is updated at every boosting iteration (see L266-268) to guarantee target detection rate. (22) is updated after each bootstrap cycle, which produces new estimate for cascade FP rate, fp^t (see L313-316).

- Conclusion not very useful
o We will update the conclusion.

- \beta update
o \beta is \in [0,1] so [ (1+ \beta)/ 2 ]will increase \beta.

R3:
- Other methods such as deep networks (DNs)
o Note that goal here is real-time detection of multiple objects in a large image on low complexity devices. This is important for many applications, e.g. driver assistance systems, where limited computation is available and recognition must control vehicle in real-time. E.g., a system that scans an image to localize all possible pedestrians and must classify millions of candidate regions per image. This is different from single patch classification, as in ImageNet, where DNs can be used. DNs, even when implemented in GPU, are not fast enough (see for example (*) which requires at least 1.5GB RAM and takes about 13s per image on a GPU or 53s per image on a CPU)

- (*)Ross Girshick, et al. “Rich feature hierarchies for accurate object detection and semantic segmentation Tech report”